# Association between the Clinical, Laboratory and Ultrasound Characteristics and the Etiology of Peripheral Lymphadenopathy in Children

**DOI:** 10.3390/children10101589

**Published:** 2023-09-23

**Authors:** Vojko Berce, Nina Rataj, Maja Dorič, Aleksandra Zorko, Tjaša Kolarič

**Affiliations:** 1Department of Pediatrics, University Medical Centre Maribor, Ljubljanska ulica 5, 2000 Maribor, Slovenia; nina.rataj@ukc-mb.si (N.R.); maja.doric@ukc-mb.si (M.D.); aleksandra.zorkobrodnik@ukc-mb.si (A.Z.); 2Community Health Center Velenje, Vodnikova cesta 1, 3320 Velenje, Slovenia; tjasa.kolaric@zd-velenje.si

**Keywords:** peripheral lymphadenopathy, children, etiology, infectious mononucleosis, cat scratch disease, bacterial lymphadenitis, ultrasound

## Abstract

Peripheral lymphadenopathy affects most children at least once in a lifetime and represents a major reason for concern. Therefore, we aimed to identify the most common causes of peripheral lymphadenopathy in hospitalized children and to determine the clinical, laboratory and ultrasound characteristics that enable fast, easy and accurate etiological diagnosis. We performed a cross-sectional study including 139 children who were hospitalized because of peripheral lymphadenopathy. Ultrasound of lymph nodes was performed in 113 (81.3%) patients. Lymphadenopathy was generalized in nine (6.5%) patients. Malignant etiology was established in only three (2.2%) patients. Bacterial lymphadenitis, infectious mononucleosis (IM) and cat scratch disease (CSD) were diagnosed in 66 (47.5%), 31 (22.3%) and 29 (20.9%) patients, respectively. Bacterial lymphadenitis was significantly associated with neutrophilia (*p* < 0.01), and increased C-reactive protein levels (*p* < 0.01). IM was associated with pharyngitis (*p* < 0.01), leukocytosis without neutrophilia (*p* = 0.03) and increased blood liver enzyme levels (*p* < 0.01). CSD was associated with recent contact with a cat (*p* < 0.01), absence of a fever (*p* < 0.01) and normal white blood cell count (*p* < 0.01). Thorough history and clinical examination in combination with a few basic laboratory tests enable fast and accurate differentiation between the most common etiologies of lymphadenopathy in children to avoid unnecessary procedures and hospitalizations.

## 1. Introduction

Peripheral lymphadenopathy is one of the most common medical conditions and a major reason for concern for children and their caregivers. It affects most children at least once a lifetime, most commonly between the ages of 4 and 8 years. Lymphadenopathy is defined as an abnormality in the size and/or consistency of lymph nodes, while the term lymphadenitis refers to lymphadenopathy that occurs from infectious and other inflammatory processes and is characterized by tender lymph nodes. However, the terms “lymphadenitis” and “lymphadenopathy” are often used interchangeably. Localized lymphadenopathy is defined as an abnormality of lymph nodes in only one region (e.g., cervical, inguinal, axillary) and can be unilateral or bilateral. Lymphadenopathy is generalized when lymph nodes are affected simultaneously in two or more noncontiguous regions [1,2].

Acute (develops over days) bilateral cervical lymphadenitis is the most common presentation of lymphadenopathy in children, and lymph nodes in the anterior cervical area are usually affected, most commonly due to viral infections or other benign conditions. Viral infections can also present as generalized lymphadenopathy [3,4,5,6]. Epstein–Barr virus (EBV) and cytomegalovirus (CMV) cause infectious mononucleosis (IM), characterized by pharyngitis and subacute (development over weeks) bilateral anterior cervical lymphadenopathy or generalized lymphadenopathy [7]. Lymphadenitis associated with Group A streptococcal (GAS) pharyngitis is also a common cause of acute bilateral cervical lymphadenopathy. Acute unilateral cervical lymphadenitis is less common than bilateral disease and is usually caused by pyogenic bacteria such as *Staphylococcus aureus* or GAS. Localized bacterial lymphadenitis can also arise as a response of the draining nodes to local infection in axillary, epitrochlear or inguinofemoral regions [6,8,9].

Granulomatous diseases such as cat scratch disease (CSD), caused by *Bartonella hensleae*, tuberculosis, and infections with nontuberculous atypical mycobacteria are common causes of subacute or chronic (development over weeks to months) localized lymphadenopathy. CSD can also rarely present with generalized lymphadenopathy [10,11].

The prevalence of malignancy among children with lymphadenopathy managed in the primary care setting is approximately 5%; however, it increases to 13–49% among children with suspected malignancy who underwent biopsy in pediatric referral centers [10,12,13]. The risk of malignancy increases with worrisome features such as systemic (B) symptoms (fever > 1 week, weight loss > 10%, night-time sweating), fixed and non-tender lymph nodes, lymph nodes > 2 cm in diameter and not responding to antibiotic therapy, unusual locations (e.g., supraclavicular, mediastinal), generalized lymphadenopathy and unexplained abnormal laboratory results such as bicytopenia or pancytopenia, persistently elevated erythrocyte sedimentation rate (ESR) and highly elevated lactate dehydrogenase (LDH) levels [4,10,14,15].

No investigation is needed in children with acute bilateral cervical or generalized lymphadenopathy associated with evident viral infection. Even unexplained localized lymphadenopathy without other risk factors for malignancy (see above) can be safely managed by two to three weeks of observation with or without antibiotic treatment [2,3,16].

Blood tests, such as complete blood count (CBC) with differential, erythrocyte sedimentation rate and C-reactive protein (CRP), are commonly performed in children with lymphadenopathy [2,17]. Ultrasound (US) is useful for differentiating lymphadenopathy from other causes of neck or groin lump and for diagnosing abscess formation in bacterial lymphadenitis. The accuracy of US to differentiate malignant from reactive lymph nodes is limited [18,19].

The gold standard for the etiological diagnosis of lymphadenopathy is an excisional biopsy of the lymph node, which is usually performed in persistent or progressive lymphadenopathy after 4–6 weeks. Biopsy is performed much earlier in children with unexplained lymphadenopathy and worrisome features. Fine needle aspiration biopsy with cytological and microbiological analysis can be performed instead of excision [2,4,20].

The most common etiologies of pediatric lymphadenopathy differ according to the world region [12]. To the best of our knowledge, there have been no such etiological studies recently performed in Central Europe.

The aim of our study was to identify the most common etiologies of peripheral lymphadenopathy among hospitalized children in our region. In addition, we wanted to assess the usefulness of clinical, laboratory and ultrasound characteristics to quickly, easily and accurately differentiate between the most common causes of lymphadenopathy in children. This differentiation is crucial for minimizing unwarranted medical investigations, demanding procedures and hospitalizations, and for initiation of appropriate antibiotic treatment when warranted.

We hypothesized that IM, CSD and bacterial lymphadenitis are the most common causes of peripheral lymphadenopathy in hospitalized children. We also hypothesized that IM is associated with higher age, bilateral cervical or generalized lymphadenopathy, pharyngitis, the presence of atypical lymphocytes, higher LDH levels and elevated blood liver enzyme levels. CSD is expected to be associated with rural living, recent contact with a cat, larger lymph nodes and localized lymphadenopathy of longer duration. In bacterial lymphadenitis, we expect the presence of fever, signs of inflammation and abscess formation (fluctuant lymph nodes), higher white blood cell (WBC) and neutrophil counts and higher CRP values than in other etiologies.

## 2. Materials and Methods

### 2.1. Participants

We conducted a cross-sectional study including all children aged from 1 month to 18 years who were admitted to our Department of Pediatrics, University Medical Centre Maribor, Slovenia from 1 January 2016 to 31 December 2021 because of enlarged peripheral lymph nodes.

We included all patients with a referral diagnosis of lymphadenitis or lymphadenopathy, regardless of the duration of the disease. We also included children who were referred with other diagnoses (e.g., tumor, lump, swelling) and who were diagnosed with lymphadenopathy as a cause of swelling during hospitalization. Only patients with enlarged peripheral lymph nodes were included. We did not include children with enlarged lymph nodes inaccessible to palpation (e.g., in the thoracic or abdominal cavity). We considered lymph nodes pathologically enlarged if their size (assessed by palpation) exceeded 1.5 cm in the inguinal region or 1 cm in the cervical, axillary, or femoral region. If the lymph nodes were palpable in any other region (e.g., supraclavicular or epitrochlear), they were considered pathological, irrespective of their size, and the patient was included in the study [17,21]. Epidemiological and clinical data such as age, sex, area of residence (urban or rural), recent close contact with a cat, duration of illness, presence of fever (at least 38 °C, tympanic or axillary) or other B symptoms (weight loss > 10%, night sweats) and presence of pharyngitis were recorded. The location and size of the lymph nodes as well as overlying skin redness and abscess formation (fluctuation) were assessed clinically.

We excluded children with previously known conditions potentially affecting lymph nodes, such as (already known) malignant, autoimmune or storage diseases. We also excluded patients with atopic dermatitis or skin infections, patients with autoinflammatory diseases such as periodic fever with aphthous stomatitis, pharyngitis and adenitis (PFAPA) and other patients in whom the enlarged peripheral lymph nodes were not a primary reason for their referral (e.g., bacterial pharyngitis, respiratory viral illnesses). Patients in whom alternative diagnoses were established as a cause of swelling during hospitalization (e.g., cyst, hemangioma, lymphangioma, hematoma) or in whom no cause of lymphadenopathy could be ascertained were also excluded from the study. We also excluded patients with lymphadenopathy who received antibiotic treatment for more than 24 h before admission.

### 2.2. Methods

A venous blood sample was drawn from all patients for the analysis of CBC and differential, presence of atypical lymphocytes (>10% of all WBC count), and levels of CRP, LDH and blood liver enzymes alanine and aspartate aminotransferase. Blood liver enzymes and LDH levels were evaluated using reference values for children [22]. Determination of specific immunoglobulin M and G class (IgM and IgG) antibody levels for CMV and EBV was performed using enzyme-linked fluorescent assay (ELFA) and enzyme-linked immunosorbent assay (ELISA), respectively [23,24]. For the diagnosis of CSD, we determined the presence of IgM and IgG to *B. hensleae* with an indirect immunofluorescence assay (IFA) [25]. Serologic testing for other pathogens (e.g., *Francisella tularensis*, *Toxoplasma gondii*, etc.) was performed when clinically appropriate and at the discretion of the treating physician. Polymerase chain reaction (PCR)-based diagnostics for *B. hensleae* or nontuberculous mycobacteria (tissue sample) and CMV (blood sample) were seldom performed. Incision and drainage were performed in 29 (20.9%) of our patients (when suppurative lymphadenitis was expected), and the sample was cultivated for bacteria. Fine needle aspiration biopsy or extirpation of lymph nodes with cytological or histological examination was performed in 17 (12.2%) patients. CBC with differential, CRP and biochemistry tests were performed in the Central Laboratory of the University Medical Center Maribor, Slovenia. Serological, microbiological (culture) and PCR testing were performed in the National Laboratory of Health, Environment and Food, Maribor, Slovenia. Cytological and histological examinations were performed in the Department of Pathology, University Medical Center Maribor, Slovenia.

US of lymph nodes was performed in 113 (81.3%) patients with the US machine Toshiba Applio 400 (Canon Medical Systems, Tokyo, Japan) using the PLT-805AT Linear Probe. The position and number of enlarged lymph nodes were recorded as well as the longitudinal diameter of the largest node and the presence of anechoic and nonperfused areas representing probable abscess formation.

### 2.3. Ethical Approval and Data Availability

This study was approved by the Ethics Committee of the University Clinical Centre Maribor (UKC-MB-KME-5/22, issued on 15 March 2022) and was conducted in accordance with the Helsinki Declaration of 1975, as revised in Edinburgh in 2000. All participants or their legal guardians (for children under 16 years of age) signed an informed consent form. The raw data used in this study are openly available in Kaagle at doi: 10.34740/kaggle/dsv/6257640.

### 2.4. Stratification of Patients

For further (inferential) statistical analysis, patients were stratified into the three most common etiological groups: infectious mononucleosis, cat-scratch disease and bacterial lymphadenitis. Patients were classified at the end of hospitalization, and the final diagnosis was once more revised by two senior consultant pediatricians. Other causes of lymphadenopathy (malignant disease, infection with nontuberculous mycobacteria, Kawasaki disease, tularemia, nonspecific viral illness, toxoplasmosis) were excluded from further statistical analysis.

The detection of CMV-specific IgM and IgM-to-EBV viral capsid antigen (VCA) in the appropriate clinical setting was considered proof of acute infection with CMV and EBV, respectively. PCR-based detection of the CMV genome in blood was also considered diagnostic for CMV infection [25,26]. Those patients were classified as having infectious mononucleosis.

Patients with detected IgM (regardless of titers) and/or IgG titers ≥ 1:256 to *B. hensleae* and/or positive PCR were (in the appropriate clinical setting) classified as having CSD [25].

When pyogenic bacteria (*S. aureus* or GAS) were cultivated (or detected with PCR) from samples obtained at incision or aspiration biopsy, those patients were stratified into the bacterial lymphadenitis group. Patients with acute lymphadenitis (and tender lymph nodes) that resolved on antibiotic therapy during hospitalization and in whom no other etiology could be established were also considered as having bacterial lymphadenitis even when definitive microbiological diagnostics were not performed.

The stratification of patients is presented in Figure 1.

### 2.5. Statistical Analysis

Statistical analysis was performed with IBM SPSS 26.0 software (IBM Inc., Chicago, IL, USA). A descriptive statistical analysis was performed on data from all included patients and was upgraded with inferential statistics in those 126 (90.6%) patients who were stratified into one of the three main etiological groups (IM, CSD, bacterial lymphadenitis). The Kolmogorov–Smirnov test was used to evaluate the normality of data distributions. The Mann–Whiney U test was performed to compare quantitative epidemiological, clinical, laboratory and US characteristics between the etiological groups. The association of the etiology of lymphadenitis with qualitative characteristics was analyzed using Fisher’s exact or chi-squared test. The risk, positive predictive value and negative predictive value were calculated for bacterial lymphadenitis. In addition, we used a (multinomial) regression model to compare clinical, laboratory and US characteristics of patients with the three most common etiologies, adjusted for age and sex. The α level for all tests was set to 0.05, and *p* values are presented for two-tailed tests.

## 3. Results

### 3.1. Epidemiological, Clinical, Laboratory and Ultrasound Characteristics

During the study period, a total of 194 children were admitted because of peripheral lymphadenopathy, and 139 (75.6%) of them met the inclusion criteria (Figure 1). Their ages ranged from 2 months to 18 years (median 50 months, interquartile range 53 months), and 67 (48.2%) of them were female. Only nine (6.5%) patients presented with generalized lymphadenopathy. In 31 (22.3%) patients, lymphadenopathy was bilateral (neck region in all cases), and 99 (71.2%) patients presented with localized unilateral lymphadenopathy. Of 130 patients with nongeneralized lymphadenopathy, 114 (87.7%), 6 (4.6%), 6 (4.6%) and 4 (3.1%) patients presented with enlarged lymph nodes in the cervical, axillary, supraclavicular and inguinofemoral areas, respectively. B symptoms were present in only four (2.9%) patients.

Other clinical and laboratory characteristics of patients with lymphadenopathy are presented in Table 1.

### 3.2. Etiology

The etiology of lymphadenopathy in our patients is presented in Table 2. 

Regarding the patients with malignant tumors, two cases were proven to be non-Hodgkin lymphoma, and one was proven to be Hodgkin disease. Lymphadenitis was caused by pyogenic bacteria, EBV (or CMV) and *B. hensleae* in 66 (47.5%), 31 (22.3%) and 29 (20.9%) patients, respectively. These three most common etiologies accounted for 90.6% of all cases. A comparison of epidemiological, clinical, laboratory and ultrasound characteristics between these three most common etiologies is presented in Table 3 and Table 4.

## 4. Discussion

In our study, we found that bacterial lymphadenitis (caused by pyogenic bacteria) was the most common cause of peripheral lymphadenopathy in hospitalized children, followed by CSD (*B. hensleae*) and IM (EBV or CMV). These three etiologies accounted for 90.6% of all cases. We proved malignant tumors as a cause of lymphadenopathy in only three (2.2%) patients, which is similar to the 2.7% found by Bozlak et al., but less than the 4.6% reported in a meta-analysis performed by Deosthali et al. [12,26] and much less than the 24.3% reported by Oguz et al. However, Oguz et al. analyzed children who were already referred to the pediatric oncology department under the suspicion of malignancy, while our study was performed in the general pediatric ward [14]. Infection with pyogenic bacteria was found to be the most common cause of lymphadenopathy in our study (47.5% of all cases), which is similar to the results of reports by Dajani et al. and Barton et al. for causes of cervical lymphadenitis [27,28]. In contrast to these two studies, we also included children with generalized lymphadenopathy and lymphadenopathy localized outside the neck, although 82.0% of our patients had cervical lymphadenopathy. We found IM to be the second most common cause of peripheral lymphadenopathy (22.3% of all cases). This prevalence is similar to that reported by Bozlak et al., who found EBV responsible for 27% of all infectious causes of lymphadenopathy [26]. Deosthali et al. also found EBV infection to be the second most common cause of cervical lymphadenopathy, although with a much lower prevalence of 8.9%. However, in this meta-analysis, a nonspecific benign etiology prevailed with 67.8%. In the same meta-analysis, granulomatous etiology was in fourth place with 4.1%, just behind malignancy, and tuberculosis prevailed among granulomatous diseases [12]. This finding is in strict contrast with our results, as we found that *B. hensleae* was responsible for most granulomatous lymphadenitis, and CSD accounted for 20.9% of all lymphadenopathies in our study. We found no tuberculous lymphadenitis, and only four cases (2.9%) were caused by atypical mycobacteria. This discrepancy probably reflects regional differences in the etiology of lymphadenopathy in children, as the meta-analysis performed by Deosthali et al. was mostly based on studies performed in undeveloped countries, where the prevalence of tuberculosis is still high [12]. However, even a study from Italy performed by De Corti et al. showed atypical mycobacteria was a cause of granulomatous peripheral lymphadenopathy at rates twice those of CSD [29].

Regarding the etiological diagnosis of the most common causes of lymphadenopathy, we found that patients with CSD were significantly older than patients with IM or bacterial lymphadenitis. Recent contact with a cat was also significantly associated with CSD, while fever was significantly less common in patients with CSD than in the other two main etiological groups. Patients with CSD also reported a longer duration of symptoms (median: 5 days) than those with bacterial lymphadenitis (median: 3 days) but not than those with IM. Pharyngitis was significantly more common in patients with IM (in 71.0%), although it was also observed in 40.9% of patients with bacterial lymphadenitis. Generalized (in 25.8%) or bilateral (54.8%) lymphadenopathy was significantly more common in patients with IM than in those with CSD or bacterial lymphadenitis. Bacterial lymphadenitis presented with significantly larger (median: 4 cm) lymph nodes than those of patients with IM but not larger than those of patients with CSD. Skin redness and/or fluctuation was present in nearly half of patients with bacterial lymphadenitis, in a quarter of patients with CSD and in none with IM, while lymph node tenderness was significantly less common in patients with IM (than in patients with other etiologies), although it was still reported in over 50%. B symptoms were present in two (out of three) patients with lymphoma and in two patients with IM.

The epidemiological and clinical findings in our patients are mostly consistent with those of a diagnostic algorithm for lymphadenopathy proposed by Gaddey et al., as they also considered the presence of fever and tender lymph nodes characteristic of viral or pyogenic bacterial infections, generalized lymphadenopathy for IM, recent contact with a cat and axillary (or inguinofemoral) location for CSD and redness, warmth and fluctuation of lymph nodes for bacterial lymphadenitis. However, in our study, only two (3.0%) patients with bacterial etiology presented with isolated axillary or inguinofemoral lymphadenitis, although Gaddey et al. reported that these two regions are relatively commonly affected in bacterial lymphadenitis [30]. Similar to our results, Long et al. also found axillary and inguinofemoral regions affected in only 1.1% and 5.5% of cases of bacterial lymphadenitis, respectively [31]. In contrast with Gaddey et al., we did not find that the duration of lymphadenopathy in CSD was subacute or chronic, although it was of longer duration (median: 5 days) than lymphadenitis caused by pyogenic bacteria (but not compared to IM). Furthermore, we found localized cervical lymphadenopathy as the most common presentation of CSD (in 55.2%), which is also in contrast with the results of Gaddey et al. and with those of other previous reports that found localized axillary lymphadenopathy to be the most common presentation of CSD [11,30]. In contrast to some previous studies, we also did not observe larger lymph nodes (than those with bacterial lymphadenitis) in our patients with CSD. However, the data regarding the absence of fever (in 69.0%), localized unilateral lymphadenopathy (in 93.1%) and lymph node tenderness (in 79.3%) in our patients with CSD were all in concordance with those of previous reports [11,30,32]. When we consider epidemiological data, such as contact with a cat, living in rural areas and the absence of leukocytosis and/or increased CRP, our results indicate that the probable diagnosis of CSD can be made even without demanding microbiological investigations or ultrasound, especially when the risk of malignancy is otherwise low.

A quarter of our patients with IM presented with generalized lymphadenopathy, which is similar as reported previously. However, almost a quarter of our patients with IM also presented with unilateral cervical lymphadenopathy. This is far more than expected from previous studies that found bilateral cervical lymphadenopathy in almost all patients with EBV-related lymphadenopathy [33,34]. Interestingly, our patients with IM were younger (median age of 45 months) than those with CSD (median age of 78 months) and were of similar age to patients with lymphadenitis caused by pyogenic bacteria. This finding is also in strict contrast with those of previous epidemiological studies, which reported that IM most commonly affects adolescents and young adults [7,34]. A probable explanation for this discrepancy is that older children and adolescents with IM do not need hospitalization and that we considered all patients with EBV-caused lymphadenitis to have IM.

Regarding the laboratory results, we found leukocytosis in patients with bacterial lymphadenitis and IM (median: 15.1 × 10^9^/L and 15.9 × 10^9^/L, respectively). However, the neutrophil count was increased only in bacterial lymphadenitis (median 9.7 × 10^9^/L) and normal in IM patients (median: 4.0 × 10^9^/L). Similarly, CRP levels were increased in bacterial lymphadenitis (median: 34 mg/L) and only marginally increased in CSD and IM patients (median: 9 mg/L and 14 mg/L, respectively). Atypical lymphocytes (>10% of WBC count) were present in 80.1% of our patients with IM and in none with other etiologies, although CSD was reported previously to also be associated with atypical lymphocytes. However, for the evaluation of peripheral blood smears, an experienced reviewer is needed, who is often not available in the primary and outpatient settings [35]. LDH levels were increased in over 90% of our patients with IM and in only 5% of children with bacterial lymphadenitis or CSD. Similarly, elevated blood liver enzyme levels were observed in 65% of patients with IM and in less than 5% of patients with bacterial lymphadenitis or CSD. Therefore, the laboratory results in our patients with EBV infection are in concordance with those of previous studies and reports [34].

Ultrasound was not performed routinely in our patients and was used mostly to detect the presence of abscess formation. Suppuration was found in 31.6% of our patients with bacterial lymphadenitis and in 14.3% of patients with CSD, which is in accordance with those of previous reports [11]. The relatively low percentage of suppuration that we detected in patients with lymphadenitis caused by pyogenic bacteria can be explained by the short duration of symptoms before the hospitalization and administration of antibiotics at admission in most patients (when IM was excluded). US-measured lymph nodes were smaller than those evaluated clinically. This difference can be explained by US findings of multiple enlarged lymph nodes in all patients with IM and in most patients with CSD and bacterial lymphadenitis. Clinically, a lump of multiple enlarged lymph nodes is often considered and measured as one. The difference in size detected between the US and clinical evaluation was especially obvious in bacterial lymphadenitis, where swelling of tissues around the lymph node can also contribute to the overestimation of its size [8]. Differences in the size of lymph nodes between the three main etiological groups were more pronounced when evaluated clinically than when measured with US.

Only one (16.6%) of our six patients who presented with supraclavicular lymphadenopathy was diagnosed with malignant disease (lymphoma), although biopsy or at least fine needle aspiration was performed in all of them. The other five (83.3%) were diagnosed with CSD. Although we included only a small number of patients with supraclavicular lymphadenopathy, our results are in contrast with those of Sen et al., who found a strong association of supraclavicular location with malignancy. However, the study was performed in a pediatric oncology department where only selected patients were referred, and the percentage of malignancy in patients with lymphadenopathy was found to be 7.8%, which is much higher than the 2.2% in our study [36]. Therefore, CSD should be considered a possible cause of supraclavicular lymphadenopathy in pediatric patients, especially when the history of cat scratching is positive and after the exclusion of malignant disease.

Our study exhibits several limitations. First, we excluded quite a few children in whom no etiological diagnosis of lymphadenopathy could be ascertained. Second, the diagnosis of bacterial lymphadenitis was not established with the detection of bacteria (e.g., cultivation, PCR) in all cases because there was no need for incision or needle aspiration in approximately half of our patients with presumed bacterial etiology, and those patients were stratified into the bacterial lymphadenitis group according to criteria presented in the Section 2.2 and Figure 1. Third, we included only a few patients with malignant etiology; therefore, our predictive model cannot be used to rule out cancer.

## 5. Conclusions

A thorough history and clinical examination, complemented by a few basic laboratory tests, allow a rapid and accurate diagnosis in most children with acute peripheral lymphadenopathy. However, we should always actively search for and take into account any features of concern for malignancy. Such an approach avoids unnecessary hospitalizations and demanding and costly investigations and is a prerequisite for rational antibiotic treatment.

## Figures and Tables

**Figure 1 children-10-01589-f001:**
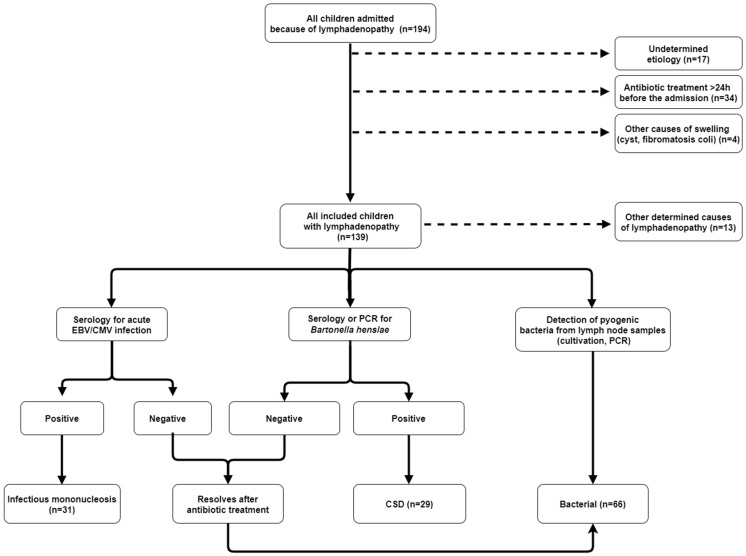
Stratification of patients with peripheral lymphadenopathy. EBV—Epstein-Barr virus; CMV—cytomegalovirus; CSD—cat scratch disease; PCR—polymerase chain reaction.

**Table 1 children-10-01589-t001:** Epidemiological, clinical, laboratory and ultrasound characteristics of children with lymphadenopathy.

Qualitative Characteristic	Frequency (N)	Percentage of Patients
Fever	88	63.3
Recent contact with a cat	48	34.5
Residence in a rural area	76	54.7
Generalized lymphadenopathy	9	6.5
Bilateral lymphadenopathy	31	22.3
Inflammation (redness and/or fluctuation)	31	22,3
Lymph node tenderness	105	75.5
Pharyngitis	60	43.2
Increased blood liver enzyme levels ^1^	24	17.3
Atypical lymphocytes (>10% of differential WBC count)	25	18.0
US ^2^, multiple enlarged lymph nodes	95	84.1
US ^2^, abscess formation	27	23.9
**Quantitative characteristic**	**Median**	**Interquartile range**
Age (months)	50	53
Duration of lymphadenopathy (days)	4	5
Size of lymph node (clinically-cm)	3	3
WBC count (×10^9^/L)	13.4	9.1
Neutrophil count (×10^9^/L)	7.1	7.0
CRP (mg/L)	20	43
LDH (μkat/L)	4,8	1.8
Size of lymph node (US ^2^-cm)	2.5	1.0

^1^ Blood liver enzyme levels were evaluated using reference values for children [24]. ^2^ Ultrasound of lymph nodes was performed in 113 (81.3%) patients, and the longitudinal diameter of the largest lymph node was recorded. LDH—lactate dehydrogenase; US—ultrasound; WBC—white blood cell; CRP—C-reactive protein.

**Table 2 children-10-01589-t002:** Etiology of peripheral lymphadenopathy in children.

Etiology	Frequency (N)	Percentage of Patients
Bacterial lymphadenitis (*Stapyhlococcus aureus* or GAS)	66	47.5
Epstein-Barr virus	29	20.9
Cat scratch disease	29	20.9
Atypical mycobacteria	4	2.9
Lymphoma	3	2.2
Kawasaki disease	2	1.4
Cytomegalovirus	2	1.4
Other specified viral infections	2	1.4
Toxoplasmosis	1	0.7
Tularemia	1	0.7

GAS—Group A beta-hemolytic streptococcus.

**Table 3 children-10-01589-t003:** Comparison of epidemiological, clinical and laboratory characteristics according to the etiology of peripheral lymphadenopathy in children.

Qualitative Characteristic [*n* (%)] ^1^	CSD (*n* = 29)	IM (*n* = 31)	Bacterial (*n* = 66)	*p* Value ^2^	Odds Ratio (95% Confidence Interval) ^3^		Positive Predictive Value (%) ^4^	Negative Predictive Value (%) ^4^
Female sex	15 (51.7)	11 (35.5)	34 (51.5)	CI = 0.297 BC = 1.000 BI = 0.191	CI = 1.40 (0.83–2.36) BC = 1.00 (0.76–1.30) BI = 1.23 (0.94–1.61)		56.7	51.5
Fever	9 (31.0)	23 (74.2)	48 (72.7)	**CI = 0.002** **BC < 0.001** BI = 1.000	CI = 0.39 (0.22–0.72) BC = 1.78 (1.25–2.53) BI = 0.98 (0.72–1.32)		60.0	60.9
Recent contact with a cat	18 (62.1)	5 (16.1)	20 (30.3)	**CI < 0.001** **BC = 0.006** BI = 0.213	CI = 2.63 (1.53–4.52) BC = 0.65 (0.47–0.91) BI = 1.25 (0.96–1.63)		46.5	44.5
Residence in a rural area	18 (62.1)	21 (67.8)	30 (45.5)	CI = 0.788 BC = 0.182 BI = 0.051	CI = 0.88 (0.52–1.51) BC = 0.62 (0.33–1.27) BI = 0.53 (0.28–1.00)		63.2	56.5
Generalized lymphadenopathy	1 (3.4)	8 (25.8)	0 (0)	**CI = 0.029** BC = 0.305 **BI < 0.001**	CI = 0.53 (0.37–0.77)			43.5
Inflammation (redness and/or fluctuation)	7 (24.1)	0 (0)	21 (31.8)	**CI = 0.004** BC = 0.626 **BI < 0.001**	CI = 2.41 (1.75–3.32) BC = 1.12 (0.85–1.47) BI = 1.67 (1.40–2.04)		75.0	54.1
Lymph node tenderness	23 (79.3)	17 (54.8)	58 (87.8)	CI = 0.058 BC = 0.348 **BI = 0.001**	CI = 1.92 (0.93–3.94) BC = 1.25 (0.78–2.01) BI = 2.13 (1.21–3.75)		59.2	71.4
Pharyngitis	4 (13.8)	22 (71.0)	27 (40.9)	**CI < 0.001** **BC = 0.009** **BI = 0.006**	CI = 0.21 (0.08–0.53) BC = 1.43 (1.13–1.82) BI = 0.68 (0.51–0.90)		50.9	46.6
Increased blood liver enzyme levels ^5^	1 (3.4)	20 (64.5)	3 (4.5)	**CI < 0.001** BC = 1.000 **BI < 0.001**	CI = 0.07 (0.01–0.45) BC = 1.08 (0.60–1.94) BI = 0.15 (0.05–0.44)		12.5	38.2
Atypical lymphocytes (>10% of differential WBC count)	0 (0)	25 (75.8)	0 (0)	**CI < 0.001** **BI < 0.001**				34.7
**Quantitative characteristic [median (IQR)]**								
Age (months)	78 (90)	45 (61)	45 (35)	CI = 0.089 **BC = 0.010** BI = 0.358				
Duration of lymphadenopathy (days)	5 (13)	3 (5)	3 (4)	CI = 0.197 **BC = 0.011** BI = 0.139				
Size of lymph node (clinically-cm)	3 (3.1)	3 (2)	4 (2)	CI = 0.945 **BC = 0.043** BI = 0.019				
WBC count (×10^9^/L)	9.9 (4.9)	15.9 (9.8)	15.1 (9.3)	**CI = 0.002** **BC < 0.001** BI = 0.634				
Neutrophil count (×10^9^/L)	5.6 (4.5)	4.0 (3.7)	9.7 (7.4)	**CI = 0.030** **BC < 0.001** **BI < 0.001**				
CRP (mg/L)	9 (21)	14 (34)	34 (48)	CI = 0.124 **BC < 0.001** **BI = 0.002**				
LDH (μkat/L)	4.1 (1.6)	7.2 (4.3)	4.4 (1.0)	**CI < 0.001** BC = 1.000 **BI < 0.001**				

^1^ Number of subjects with a characteristic (percentage in parentheses). ^2^ *p* value refers to the comparison between cat scratch disease and infectious mononucleosis (CI), between bacterial lymphadenitis and cat scratch disease (BC), and between bacterial lymphadenitis and infectious mononucleosis (BI). ^3^ Odds ratio is calculated for bacterial lymphadenitis (BC and BI) or cat scratch disease (CI). ^4^ Positive and negative predictive value is calculated for bacterial lymphadenitis. ^5^ Blood liver enzyme levels were evaluated using a reference value for children [24]. IM—infectious mononucleosis; CSD—cat scratch disease; IQR—interquartile range; WBC—white blood cell; CRP—C-reactive protein; LDH—lactate dehydrogenase.

**Table 4 children-10-01589-t004:** Comparison of ultrasound characteristics according to the etiology of peripheral lymphadenopathy in children.

Qualitative Characteristic [*n* (%)] ^1^	CSD (*n* = 28)	IM (*n* = 13)	Bacterial (*n* = 60)	*p* Value ^2^	Odds Ratio (95% Confidence Interval) ^3^	Positive Predictive Value (%) ^4^	Negative Predictive Value (%) ^4^
Multiple enlarged lymph nodes	22 (78.6)	13 (100)	50 (83.3)	CI = 0.152 BC = 0.569 BI = 0.192	CI = 0.77 (0.51–1.29) BC = 1.23 (0.60–2.53) BI = 0.86 (0.57–1.39)	58.8	37.5
Abscess formation	4 (14.3)	1 (7.7)	19 (31.6)	CI = 1.000 BC = 0.118 BI = 0.097	CI = 1.20 (0.73–1.97) BC = 1.31 (1.01–1.71) BI = 1.23 (0.99–1.47)	79.8	46.8
**Quantitative characteristic [median (IQR)]**							
Size of lymph node (cm) ^5^	2.7 (1.1)	2.5 (0.8)	2.3 (0.7)	CI = 0.441 BC = 0.027 BI = 0.512			

^1^ Number of subjects with a characteristic (percentage in parentheses). ^2^ *p* value refers to the comparison between cat scratch disease and infectious mononucleosis (CI), between bacterial lymphadenitis and cat scratch disease (BC), and between bacterial lymphadenitis and infectious mononucleosis (BI). ^3^ Odds ratio is calculated for bacterial lymphadenitis (BC and BI) or cat scratch disease (CI). ^4^ Positive and negative predictive value is calculated for bacterial lymphadenitis. ^5^ Measurement refers to the longitudinal diameter of the largest lymph node. IM—infectious mononucleosis; CSD—cat scratch disease; IQR—interquartile range.

## Data Availability

The raw data used in this study are openly available in Kaagle at doi: 10.34740/kaggle/dsv/6257640.

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
