# Peer review of "Association between the Clinical, Laboratory and Ultrasound Characteristics and the Etiology of Peripheral Lymphadenopathy in Children"

_children, 2023, doi:10.3390/children10101589_

Round 1

Reviewer 1 Report

This study investigated the association between clinical, laboratory, ultrasound characteristic and the etiology of peripheral lymphadenopathy in children. Overall, the study is well-designed and the manuscript is well-written. I just have several minor suggestions.

1. Title should be revised as "Association between .... and the etiology..."

2. The introduction section is too lengthy. Part of them could be removed, such as line 42-45 and please shorten line 46-104. 

3. please rewirte the last paragraph in the introduction. Just clearly state the object of this study.

Author Response

Comment 1. Title should be revised as "Association between .... and the etiology..."

Response: Corrected as suggested.

Comment 2. The introduction section is too lengthy. Part of them could be removed, such as line 42-45 and please shorten line 46-104. 

Response: Corrected as suggested. Text in lines 42-45 was completely deleted (with two references moved and therefore renumbered, one of them replaced with a more recent one, according to the reviewers 2 suggestion) and the rest of the text between lines 46-104 was significantly shortened (removed lines: 49-52, 59-60, 66-68, 80-84 with two references therefore deleted, and 92-97)

Comment 3. please rewirte the last paragraph in the introduction. Just clearly state the object of this study. 

Response: The aims and objectives are presented in the penultimate paragraph of  the Introduction section, before the hypotheses. For their better visibility, we created a separate paragraph, dedicated solely to this issue.

We have also rewritten the whole paragraph and the text is now as follows: "The aim of our study was to identify the most common etiologies of peripheral lymphadenopathy among hospitalized children in our region. In addition, we wanted to assess the usefulness of clinical, laboratory, and ultrasound characteristics to quickly, easily, and accurately differentiate between the most common causes of lymphadenopathy in children. This differentiation is crucial for minimizing unwarranted medical investigations, demanding procedures and hospitalizations, and for initiation of appropriate antibiotic treatment when warranted."

Reviewer 2 Report

The manuscript is well done. To enhance it, I propose the following modifications:
1. Line 126. Reference to the department's location should be made (hospital, city, country).
2. The bibliography can be improved.

Author Response

Comment 1. Line 126. Reference to the department's location should be made (hospital, city, country).

Response: Added as suggested (added text is as follows: "University Medical Centre, Maribor, Slovenia")

Comment 2. The bibliography can be improved. 

Response: Three relatively outdated references were replaced by recent ones.

Reference Nr. 10  (Gosche, J.R.; Vick, L. Acute, subacute, and chronic cervical lymphadenitis in children. Semin Pediatr Surg 2006) was replaced by a very recent one (Howard-Jones, A.R.; Al Abdali, K.; Britton, P.N. Acute bacterial lymphadenitis in children: a retrospective, cross-sectional study. Eur J Pediatr 2023).

Reference No. 4 (Slap, G.B.; Brooks, J.S.J.; Schwartz, J.S. When to perform biopsies of enlarged peripheral lymph nodes in young patients. JAMA 1984) was replaced by a much more more recent one (Farndon, S.; Behjati, S.; Jonas, N.; Messahel, B. How to use… lymph node biopsy in paediatrics. Arch Dis Child Educ Pract Ed 2017).

Reference No. 15 (Knight, P.J.; Mulne, A.F.; Vassy, L.E. When is lymph node biopsy indicated in children with enlarged peripheral lymph nodes? Pediatrics 1982) has been replaced by a much more recent one (Sgro, J.M.; Campisi E.S.; Selvam, S.; Greer, M.C.; Alexander, S.; Ngan, B.; Campisi, P. Cervical lymph node biopsies in the evaluation of children with suspected lymphoproliferative disorders; Experience in a tertiary pediatric setting. J Pediatr Surg 2022) and the text to which this reference refers has been slightly modified (lines 71-72, "13-49%" instead of "13-27%" and we added the text - "with suspected malignancy" before this reference).

However, because of the shortening of the Introduction section according to the reviewer 1 suggestion, we removed two references (No. 17 - Meier et al and No. 18 - Geddes et al) and reference No. 4 (Slap et al.) which was replaced according to your suggestion by Farndon et al.) was moved (now Nr. 20) and renumbered. 

For the same reasons, we also renumbered all the other references.

Reviewer 3 Report

The subject of the article is extremely important and actual. 

I have the following comments and questions for the authors.

There are many awkward phrases that I do not point out here; I only point out those where the meaning cannot be interpreted:

FIGURE 1 is extremely hard to read.

The paragraph between 131-137 is not clear can be rewrite in more clear format.

The conclusion need to clear and specific.

My recommendation is to focus on 3 short conclusion!

Please recheck the References order.

Please double check the article by a native English reader.

Please send to a native English speaker!

Author Response

Comment: FIGURE 1 is extremely hard to read.

Response: Figure 1 has been modified by increasing the size of the letters (font) and boxes. We believe that in this way we have improved its readability.

Comment: The paragraph between 131-137 is not clear and can be rewritten in more clear format.

Response: Part of the paragraph (between lines 131 and 136) has been rewritten to make the inclusion criteria more understandable. The text is now as follows: "Only patients with enlarged peripheral lymph nodes were included. We did not include children with enlarged lymph nodes inaccessible to palpation (e.g. in the thoracic or abdominal cavity). We considered lymph nodes pathologically enlarged if their size (assessed by palpation) exceeded 1.5 cm in the inguinal region or 1 cm in the cervical, axillary, or femoral region. If the lymph nodes were palpable in any other region (e.g. supraclavicular or epitrochlear), they were considered pathological, irrespective of their size, and the patient was included in the study."

Comment The conclusion need to be clear and specific. My recommendation is to focus on 3 short conclusions!

Response: Corrected as suggested, and the whole paragraph was rearranged so that only three conclusions remained. The conclusion is now as follows:

"A thorough history and clinical examination, complemented by a few basic laboratory tests, allow a rapid and accurate diagnosis in most children with acute peripheral lymphadenopathy. However, we should always actively look for and take into account any features of concern for malignancy. Such an approach avoids unnecessary hospitalizations and demanding and costly investigations and is a prerequisite for rational antibiotic treatment."

Comment: Please recheck the References order.

Response: We checked the order of the references again and found no errors. However, we found some minor errors in the citation of the sources themselves. For example, we have corrected the page interval of reference No. 1, and we have also inserted some missing punctuation marks.

Comment: Please double check the article by a native English reader.

Response: The article was edited for proper English language, grammar, punctuation, spelling, and overall style by qualified native English-speaking editors at AJE (American Journal Expert) editing service (Nature Journals).
The Editing certificate was issued on April 1, 2022 and was provided to Children's Editorial Office. The Editing Certificate may be verified on the AJE website using the verification code CE4E-F230-172E-7B68-8AFB. However, if the Children's Editorial Office considers additional English proofreading and editing necessary, we will use the MDPI editing service.